# Pan-Cancer Analysis Reveals PPRC1 as a Novel Prognostic Biomarker in Ovarian Cancer and Hepatocellular Carcinoma

**DOI:** 10.3390/medicina59040784

**Published:** 2023-04-17

**Authors:** Xingqiu Ruan, Guoliang Cui, Changyu Li, Zhiguang Sun

**Affiliations:** 1Department of Gastroenterology, The Second Affiliated Hospital of Nanjing University of Chinese Medicine, Nanjing 210023, China; at2115@126.com (X.R.);; 2The Second Clinical Medical Collegel, Nanjing University of Chinese Medicine, Nanjing 210023, China; 3Department of Integrated Chinese and Western Medicine, Red Cross Hospital of Yulin City, Yulin 537000, China; 4Department of Rehabilitation Medicine, Red Cross Hospital of Yulin City, Yulin 537000, China; 5The First Clinical Medical College, Nanjing University of Chinese Medicine, Nanjing 210023, China

**Keywords:** PPRC1, pan-cancer, prognosis, immune checkpoint, expression

## Abstract

*Background and Objectives*: As is well understood, peroxisome proliferator-activated receptor gamma cofactor-related 1 (*PPRC1*) plays a central role in the transcriptional control of the mitochondrial biogenesis and oxidative phosphorylation (OXPHOS) process, yet its critical role in pan-cancer remains unclear. *Materials and Methods*: In this paper, the expression levels of PPRC1 in different tumor tissues and corresponding adjacent normal tissues were analyzed based on four databases: The Genotype-Tissue Expression (GTEx), Cancer Cell Line Encyclopedia (CCLE), The Cancer Genome Atlas (TCGA), and Tumor Immune Estimation Resource (TIMER). Meanwhile, the prognostic value of *PPRC1* was inferred using Kaplan–Meier plotter and forest-plot studies. In addition, the correlation between *PPRC1* expression and tumor immune cell infiltration, immune checkpoints, and the tumor-stemness index was analyzed using TCGA and TIMER databases. *Results*: According to our findings, the expression level of *PPRC1* was found to be different in different cancer types and there was a positive correlation between *PPRC1* expression and prognosis in several tumor types. In addition, *PPRC1* expression was found to be significantly correlated with immune cell infiltration, immune checkpoints, and the tumor-stemness index in both ovarian and hepatocellular carcinoma. *Conclusions*: *PPRC1* demonstrated promising potential as a novel biomarker in pan-cancer due to its potential association with immune cell infiltration, expression of immune checkpoints, and the tumor-stemness index.

## 1. Introduction

A tumor is a heterotrophic proliferative tissue, marked by locally abnormal proliferating cells that lose their regulatory effect on normal growth at the genetic level due to the activation of various carcinogenic factors and gene mutations in the organism resulting in clonal abnormal proliferation, which often manifests as a local mass. The development of a tumor is a multifactorial and multi-step process involving the proliferation, anti-apoptosis, enhanced angiogenesis, and immune escape of cancer cells. Cancer is a common malignant tumor in clinical practice, and this disease has a high mortality rate with rapid progression and causes great physical, psychological, and economic harm to patients [1,2]. Recent studies in the literature have shown that an important role in tumor-tissue microenvironment alteration is played by stromal cells and immune cells. Therefore, the discovery of new cancer biomarkers is also closely related to some key genes [3]. It is thus clear that a pan-cancer analysis not only allows the identification of key genes in different cancer types, but also attempts to summarize the diagnosis and treatment of more tumors by cross-tumor similarity [4,5], which fundamentally improves the effectiveness of cancer treatment.

The peroxisome proliferator-activated receptor γ cofactor (PGC) family is a well-known family of activated mitochondrial synthesis molecules, mainly PGC-1α (*PPARGC1A*), PGC-1β *(PPARGC1B*), and PRC (*PPRC1*) [6]. The PGC family plays a key role in regulating energy metabolism, maintaining glucose homeostasis, and lipid metabolism [7,8]. It is also a major regulator of mitochondrial oxidative energy metabolism and antioxidant defense [6]. PGC-1α can increase ATP production by regulating mitochondrial biosynthesis and energy metabolism, promoting tumor angiogenesis, and thus increasing tumor invasiveness by making it easier for cancer cells easier to proliferate, metastasize, and increase their degree of malignancy [9]. PGC-1β plays an important role in the formation of mitochondria and influences fat formation, repair of sport-related muscle injury, and regulation of vascular proliferation. Recent studies have also shown that PGC-1β affects the growth and proliferation of tumor cells such as breast cancer, liver cancer, and glioma [10,11,12]. Comparatively few studies on *PPRC1* have investigated the similar biological functions between PGC-1α and PGC-1β [13,14], while many have shown that PPRC1 not only plays an important role in improving and promoting oxidative stress, inflammation, and metabolic reprogramming but is also critical in regulating mitochondrial biogenesis and maintaining expression by coordinating the metabolic stress response [15]. It is well known that tumorigenesis is closely related to oxidative stress, metabolic reprogramming, and the regulation of mitochondrial energy metabolism. Recent literature reports have also shown that *PPRC1* not only participates in regulating glioblastoma [15,16,17] but also helps in diagnosing and understanding the pathogenesis of polymorphic low-grade neurons in young people [18]. Nevertheless, studies of the regulatory mechanism of *PPRC1* in malignancy are rare and deserve further investigation.

In summary, we hypothesized that, based on the results of comprehensive analysis of multiple databases, *PPRC1* expression was significantly correlated with tumor immune cells, immune-checkpoint genes, and tumor stemness. The results of the present study revealed that *PPRC1* may be a potentially critical biomarker suggestive of prognosis in ovarian and hepatocellular carcinoma.

## 2. Method

### 2.1. Expression Level of PPRC1

First, we screened the expression of *PPRC1* in normal tissues by the Genotype-Tissue Expression (GTEx) database and the expression of PPRC1 in tumor cell lines by the Cancer Cell Line Encyclopedia (CCLE) database. The expression of *PPRC1* in normal and tumor tissues was inferred by integrating data from three databases, TIMER, The Cancer Genome Atlas (TCGA), and GTEx [19]. The “rma function” (http://www.r-project.org/
https://www.rstudio.com/, accessed on 17 July 2022) of R software (studio version: 1.2.1335, R version: 3.6.1, Ross Ihaka, Robert Gentleman, University of Auckland, New Zealand) was used to collate the entire dataset, exclude missing and duplicate results, and to analyze the results by the log2 (TPM + 1) analysis method for the final transformation of the eligible data.

### 2.2. Prognostic Role of PPRC1

First, the Kaplan–Meier statistical method was used to analyze the survival differences between the two groups of patients. Secondly, according to the statistical method of univariate Cox regression model, the favorable or unfavorable survival outcomes of *PPRC1* were analyzed from three aspects: overall survival (OS), disease specific survival (DSS), and progression-free interval (PFI). The Cox regression forest map was formed using the R package “survivor” and “forest map”. In addition, the Kaplan–Meier plotter was used to further analyze the correlation between *PPRC1* expression and survival rate (https://kmplot.com/Analysis/, accessed on 17 July 2022) [20,21,22].

### 2.3. Correlation of PPRC1 and Tumor Immune Cells

The TIMER database allowed us to perform correlation studies between *PPRC1* and six types of immune cell infiltration, including B cells, CD4+ T cells, CD8+ T cells, neutrophils, macrophages, and dendritic cells

### 2.4. Relationship between PPRC1 with Checkpoint Genes and Tumor-Stemness Index

We first analyzed the correlation between the expression of 47 immune-checkpoint genes and the expression of *PPRC1* in the TCGA database using Pearson’s method. Secondly, we analyzed the correlation between *PPRC1* expression and tumor dryness index (DNAss) [20,21]. Previous studies have shown that the occurrence of certain diseases, such as cancer, is closely related to abnormal expression and dysfunction of immune-checkpoint molecules [22]. Using the “Tools” module on the Sangerbox website, we studied the correlation between *PPRC1* and 47 immune-checkpoint-related genes (such as *ADORA2A*, *BTLA*, *CD274*, *CD276*, and *CD200*) in 34 cancer types. In addition, the correlation between *PPRC1* expression and DNAss in each tumor was obtained by calculating the correlation test function in the R package “psych”.

### 2.5. Statistical Analysis

The R software version used in this paper was 3.6.3 (developed by Ross Ihaka, Robert Gentleman, University of Auckland, New Zealand), which includes the R packages “ggplot2”, “ggpubr”, “limma”, “survival”, “surveyor”, and “forestplot” [16,20]. The t-test was used to analyze continuous variables with normal distribution and correlation analysis using Pearson correlation analysis was mainly used to analyze correlation studies. A *p*-value < 0.05 indicated statistical significance. The statistical analysis for comparing *PPRC1* expression in tumor and normal tissues was the Wilcoxon rank sum test. The correlation of *PPRC1* expression with immune-checkpoint genes and R-immune infiltrating cells was also analyzed by Pearson correlation analysis, and a *p* value < 0.05 indicated a significant correlation.

## 3. Results

### 3.1. Expression Patterns of PPRC1 in Pan-Cancer

We analyzed the protein levels of *PPRC1* in several public databases to analyze the expression of *PPRC1* in different tumors. First, in order to demonstrate the protein expression level of *PPRC1* in different tissues of healthy subjects, we used the data from the GTEx database for comprehensive analysis to obtain the final results. As shown in Figure 1A, although *PPRC1* protein is expressed in 31 human tissues, there were differences in protein quantification of *PPRC1* in different tissue types. Secondly, we obtained the protein expression of *PPRC1* in different tumor cell lines through the CCLE database, and there were differences (Figure 1B). Next, in order to clarify whether *PPRC1* is differentially expressed in tumors, we analyzed the expression of *PPRC1* in cancer and normal tissues from the TCGA database. Due to the relatively small number of normal samples in the TCGA database, we integrated the results of the TCGA and GTEx databases to analyze the difference in *PPRC1* expression. From the TCGA results in the TIMER database, we found that *PPRC1* was differentially expressed in 12 of 33 tumors, but when the GTEx database contained more normal samples, it was observed that *PPRC1* was differentially expressed in 27 of 33% of tumors (Figure 1C,D). These results indicate that *PPRC1* has different expression in different tumors, indicating that *PPRC1* plays an important role in the occurrence and development of tumors.

### 3.2. Prognostic Role of PPRC1 in Pan-Cancer

To further analyze the prognostic value of *PPRC1* for survival in pan-cancer, we first analyzed the correlation between *PPRC1* expression and different survival outcomes in various cancers, such as OS, DSS, and PFI. Survival analysis showed that high *PPRC1* expression in six tumors, including Adrenocortical carcinoma (ACC), Bladder Urothelial Carcinoma (BLCA), Kidney renal papillary cell carcinoma (KIRP), Liver hepatocellular carcinoma (LIHC), Ovarian serous cystadenocarcinoma (OV), and Skin Cutaneous Melanoma(SKCM), was associated with poorer OS (Figure 2A–F). The forest plots suggest that the hazard ratios (HRs) of *PPRC1* in pan-cancer and *PPRC1* in the six cancer types were significantly different, for example, *ACC* (HR = 1. 07, *p* < 0.0001), *BLCA* (HR = 1.01, *p* = 0.030), *KIRP* (HR = 1.05, *p* = 0.0015), *LIHC* (HR = 1.02, *p* = 0.0220), *OV* (HR = 1.01, *p* = 0.0420), and *SKCM* (HR = 1.01, *p* = 0.0075) (Figure 2G).

As for DSS, tumor patients with lower *PPRC1* expression were found to have better survival outcomes in ACC, KIRP and LIHC, while higher *PPRC1* was correlated with better DSS in Esophageal carcinoma(ESCA)patients (Figure 3A–D). The forest plot showed that *PPRC1* was associated with DFI in 4 tumors: ACC (HR = 1.07, *p* = 0.0020), ESCA (HR = 0.98, *p* = 0.0410), KIRP (HR = 1.06, *p* = 0.0410), and LIHC (HR = 1.02, *p* = 0.0250) (Figure 3E).

Based on Kaplan–Meier plotter database, we detected the relationship between *PPRC1* expression and different stage of LIHC and OV. In OV, a higher expression level of *PPRC1* was associated with poor prognosis at stages 2 and 4, while in LIHC, a higher expression level of *PPRC1* was correlated with poor prognosis at stage 3 (Table 1).

The PFI results showed that high expression of *PPRC1* was significantly correlated with six cancer types, with a positive correlation with GBM and a negative correlation with ACC, BLCA, KIRP, LIHC, and UVM (Figure 4A–F). The results of univariate Cox regression analysis suggested that, PPRC1 was a risk factor for ACC (HR = 1.08, *p* < 0.0001), BLCA (HR = 1.01, *p* = 0.0460), KIRP (HR = 1.05, *p* = 0.0011), LIHC (HR = 1.03, *p* = 0.0013), and UVM (HR = 1.05, *p* = 0.0310) (Figure 4G).

Immunohistochemistry is a cell biological technique that utilizes the highly specific characteristics of the binding of antibodies and antigens to ultimately detect the location and expression of a certain protein in cells. The Human Protein Atlas (HPA) (https://www.proteinatlas.org, accessed on 17 July 2022) is the largest and most complete atlas of egg-protein expression in all major tissues and organs of the human body. It provides in-depth information on the location and expression of all characteristic egg-like proteins from seven different levels, including tissue specificity, drug resistance, carcinogenicity, and regulation. Through the HPA database we found differences in *PPRC1* expression between normal tissues and ovarian cancer and liver cancer tissues. IHC results showed that the protein expression of *PPRC1* was upregulated in ovarian cancer and liver cancer tissues compared to normal tissues. These results are consistent with the results predicted by our data.

Based on the high specificity of binding between antibodies and antigens, immunohistochemistry can reveal the relative distribution and abundance of proteins. The Human Protein Atlas (HPA) (https://www.proteinatlas.org, accessed on 17 July 2022) is the largest and most comprehensive human tissue, cell, and protein spatial distribution database, with enormous capabilities and potential application prospects. We used HPA database to observe the differences in *PPRC1* expression between normal tissues and ovarian cancer and liver cancer tissues. IHC results showed that the protein expression of *PPRC1* in ovarian cancer and liver cancer tissues were upregulated compared to normal tissues. These results confirmed our findings (Appendix A).

Using the TIMER database, we investigated the correlation between the *PPRC1* expression and mutation status. The results showed that the tumors with the highest frequencies of *PPRC1* mutations were UCE, SKCM, and COAD, while the frequency of *PPRC1* mutations in OV and LIHC were 0.7% and 0.5%, respectively (Appendix A).

We analyzed the role of PPRC1 in the prognosis of pan-cancer using Kaplan–Meier plotter and univariate Cox regression statistical methods. The above results suggest that in LIHC, ACC, and OV *PPRC1* can be a potentially important prognostic factor.

### 3.3. Correlation of PPRC1 and Immune Cells

Since immune cells in the tumor microenvironment are an important factor in targeting immunotherapy [23,24], we focused on exploring the correlation between *PPRC1* expression and immune cells in pan-cancer. Our results showed that the proportion of tumor immune cells (B cells, CD4+ T cells, CD8+ T cells, neutrophils, macrophages, and dendritic cells) was significantly correlated with *PPRC1* expression in 33 out of 38 tumor types. At least four immune cell ratios were significantly correlated with PPRC1 expression in 14 tumors, including LIHC, THCA, PCPG, PRAD, KIPAN, PAAD, KIRP, KICH, KIRC, SKCM-M, SKCM, BLCA, GBMLGG, and OV. In particular, we found a significant correlation between the ratio of immune cells and PPRC1 expression in LIHC, THCA, KIRP, and KICH, and OV, KIRP, KICH, and OV, PPRC1 expression had a positive correlation with the proportion of immune cells, while in SKCM-M and SKCM it had a negative correlation (Figure 5).

### 3.4. Relationship between PPRC1 Expression and Immune-Checkpoint Genes

It is well known that immune checkpoints play an important immunomodulatory function in tumors by suppressing the immune system, maintaining immune tolerance to human autoantigens, and regulating the functional state of target cells through corresponding ligand interactions [25,26]. In the current study, we screened the correlation between the expression of 47 immune-checkpoint genes and *PPRC1*. Our results showed a significant positive correlation between *PPRC1* expression and the expression of 47 immune-checkpoint genes, except for UCEC, TGCT, SKCM, and SARC (Figure 6). Especially in OV and LIHC, *PPRC1* expression was also positively correlated with the expression of most immune-checkpoint genes. Taken together, it can be seen that *PPRC1* can provide some guidance for tumor immunotherapy and promote personalized treatment of pan-cancer.

### 3.5. Relationship between PPRC1 Expression and Tumor-Stemness Score

Tumor stem cells are known to promote tumor progression, recurrence, metastasis, and drug resistance [27]. DNA methylation results were used to calculate the tumor-stemness index, called DNAss. Pearson analysis showed that the expression of *PPRC1* was negatively correlated with DNAss of OV, GBMLGG, and LIHC, while it was positively correlated with DNAss of TGCT, UVM, and THYM (Figure 7).

## 4. Discussion

Firstly, through comprehensive analysis of data information from multiple databases, we found that the expression level of *PPRC1* in different tumors is different. Secondly, combined with clinical-information data, we speculated that the expression of *PPRC1* is negatively correlated with survival outcomes in ovarian cancer (OV) and liver hepatocellular carcinoma (LIHC) in terms of OS, DSS, and PFI. We found that the expression of *PPRC1* is closely related to the clinical stages of LIHC and OV, indicating that screening for potential biomarkers has important clinical significance in guiding the prognosis of clinical stages. IHC staining can be used to verify the correlation between clinical stages and *PPRC1* protein-expression levels. The HPA database is based on proteomics, transcriptomics, and systems biology data, which extensively includes information on protein expression in various tumors and normal tissues. Therefore, we compared the differences in *PPRC1* expression among normal tissue, ovarian cancer, and cancer using the HPA database. IHC results showed that *PPRC1* protein expression levels were upregulated in ovarian cancer and cancer tissues compared to normal tissues. These data results are consistent with our predicted data results. To confirm that the sequence of *PPRC1* in different cancers was complete, we explored the relationship between *PPRC1* expression and mutation status through the TIMER database. The results suggested that the tumors with the highest frequency of *PPRC1* mutations are UCE, SKCM, and COAD. The frequency of *PPRC1* mutations in OV and LIHC was only 0.7% and 0.5%. It can be seen that the expression of *PPRC1* is related to the difference in the level of immune-cell infiltration. Our results even showed that the difference in *PPRC1* expression was significantly correlated with the infiltration levels of CD4+ T cells, CD8+ T cells, B cells, DC, and macrophages. In particular, in LIHC and OV, the expression of *PPRC1* is positively correlated with the proportion of DC, macrophages, and neutrophils, while in SKCM-M and SKCM, it is negatively correlated. In addition, in the tumor microenvironment, we found that PPRC1 expression in LIHC and OV had the same trend as some immune-cell molecular markers such as CD200, CD200R1, and LAIR1, although, in SKCM we found that *PPRC1* expression had the opposite trend to that of some immune cell molecular markers. The underlying reason for these phenomena is the differential enrichment patterns in the tumor microenvironment. Finally, the heterogeneity of *PPRC1* expression in different tumor types suggests that it is oncogenic, a result consistent with some previous reports in the literature [28,29,30].

In addition, there is a significant positive correlation between the expression of PPRC1 and tumor immunity in OV and LIHC. This is fundamentally due to the fact that CD4+ T cell-mediated cellular dynamics and membrane receptors can stimulate the antitumor activity of cytotoxic T lymphocytes (CTLs) [31,32]. At the same time, the combination of macrophage aggregation, anti-inflammatory macrophage reduction, and pro-inflammatory (anti-tumor) macrophage increase can play a role in tumor immunotherapy [33]. The key links in the occurrence and development of tumors are carcinogenic mutations, proliferation, and functional enhancement of tissue precursor cells (such as tissue damage and regeneration), as well as the establishment of a proinflammatory environment for tumors. Because neutrophils are an important immune component in inflammatory reactions, they play a key role in the occurrence and development of tumors. A large amount of literature has pointed out that neutrophils have a cancer-promoting effect, and they also directly support tumor-cell proliferation through various paracrine signaling pathways. In the tumor microenvironment, neutrophils also exhibit a high degree of plasticity, and together, this drives the cancer-promotion mechanism. Recently, research on the tumor microenvironment has found that there is an interaction between cancer and neutrophils. In addition, another key function of neutrophils is their ability to influence the behavior of other immune cells; inhibiting the anticancer activity of other immune cells is also one of their tumor-promoting functions [34]. Dendritic cells are a heterogeneous group of white blood cells composed of different subpopulations, which not only drive specific types of immune response, but also initiate and regulate adaptive immune responses, thereby exerting an anti-tumor immune response [35]. The expression level of *PPRC1* can predict the survival time of patients, and can also infer the infiltration of immune cells in OV and LIHC [36]. Therefore, it is clear that only by fully regulating the expression level of PPRC1 can we indirectly regulate the anti-tumor immune response, thereby exerting an immunotherapeutic effect [37]. Although immunotherapy has been recognized as one of the most promising tumor-treatment methods, its efficacy varies depending on the type of tumor [38]. Therefore, screening new tumor immune markers and potential therapeutic targets is of importance in the anti-tumor field.

Cancer stem cells are not only a class of tumor cell that can self-renew and produce heterogeneity, but they also play an important role in tumor formation, proliferation, metastasis, and recurrence [39,40]. A dryness index based on methylation data is known as DNAss. When the stem-cell index approaches 1, it suggests that cell differentiation tends to decrease and stem-cell characteristics tend to intensify. In the present study, *PPRC1* expression was found to be negatively correlated with DNAss in most tumors, especially in OV, GBM, and LIHC, suggesting that low *PPRC1* expression corresponds to a strong tumor cell stem, which also suggests easy promotion of tumor proliferation and metastasis. In addition, the negative correlation between *PPRC1* expression and DNAss has an important predictive role in determining the efficacy of immunosuppressive therapy.

Unfortunately there were still some shortcomings in this study. Although we used various bioinformatics databases to speculate on the close relationship between *PPRC1* and tumor immune cells, there was no way to carry out more experiments to verify this hypothesis. Therefore, future research should focus more on using different experimental methods to verify the diagnostic and prognostic potential of *PPRC1* in many cancers. Secondly, although the prognostic value of *PPRC1* has been plausibly explained, how *PPRC1* exerts a mechanism to regulate immune viability has not been clarified, and this requires more detailed studies at a later stage. In addition to this, further validation of our preliminary screening results at the molecular and clinical levels is needed to clarify the importance of *PPRC1* in pan-cancer.

## 5. Conclusions

Generally speaking, the results of this study have led to the important conclusion that differences in *PPRC1* expression in different malignant tumors seriously affect the survival time of tumor patients as *PPRC1* expression is upregulated in a variety of cancers and is associated with poorer prognosis. In addition, the results suggest that the expression of *PPRC1* may regulate tumor immunity in OV and LIHC. In the future, research on PPRC1 expression and tumor immune microenvironment may help to provide clear answers and provide immune-based anti-cancer strategies.

## Figures and Tables

**Figure 1 medicina-59-00784-f001:**
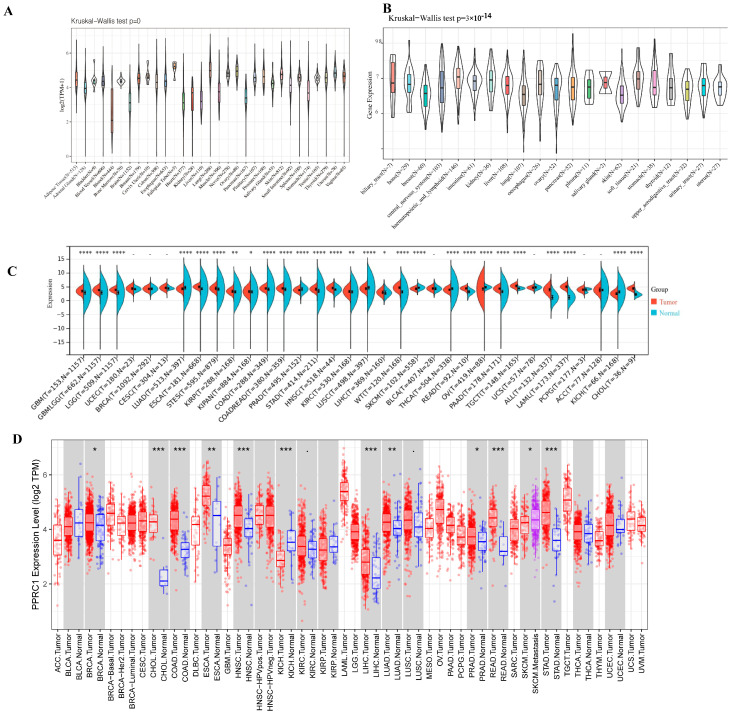
Pan-cancer expression of *PPRC1*. The expression level of *PPRC1* in normal tissues from the GTEx database (**A**) and different tumor cell lines in the CCLE database. The expression of *PPRC1* in normal and tumor tissues was explored in integrated datasets containing TCGA (**B**) and GTEx datasets (**C**), and in the TIMER database (**D**). * *p* < 0.05; ** *p* < 0.01; *** *p* < 0.001, **** *p* < 0.0001.

**Figure 2 medicina-59-00784-f002:**
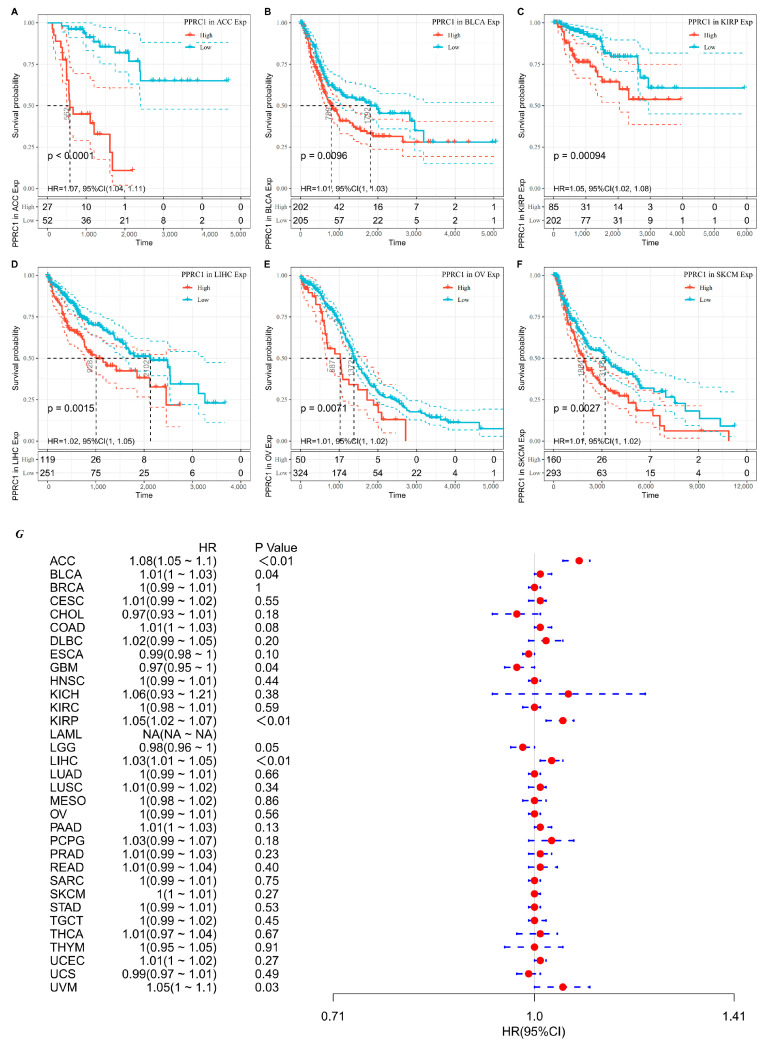
The association of *PPRC1* with OS. Kaplan–Meier-plotted survival analysis of different expression levels of *PPRC1* from the TCGA database. The OS of ACC (**A**), BLCA (**B**), KIRP (**C**), LIHC (**D**), OV (**E**), and SKCM (**F**) in patients with higher or lower *PPRC1* expression. The forest plot shows the hazard ratios of PPRC1 in pan-cancer (**G**).

**Figure 3 medicina-59-00784-f003:**
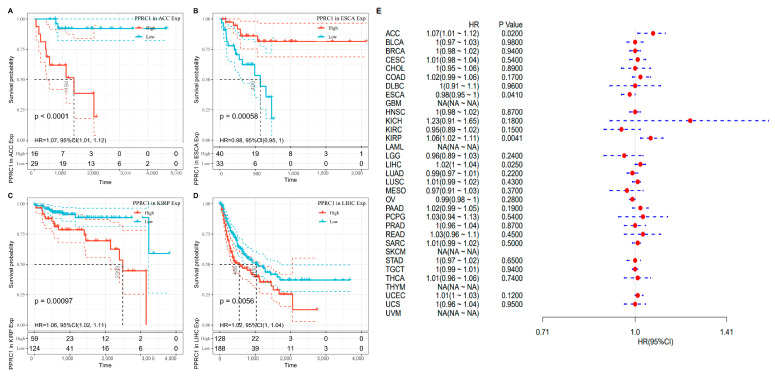
The association of *PPRC1* with DSS. Kaplan–Meier-plotted survival analysis revealed the DSS of ACC (**A**), ESCA (**B**), KIRP (**C**), and LIHC (**D**) in patients with different expression levels of *PPRC1*. The forest plot showed the hazard ratios of *PPRC1* in pan-cancer (**E**).

**Figure 4 medicina-59-00784-f004:**
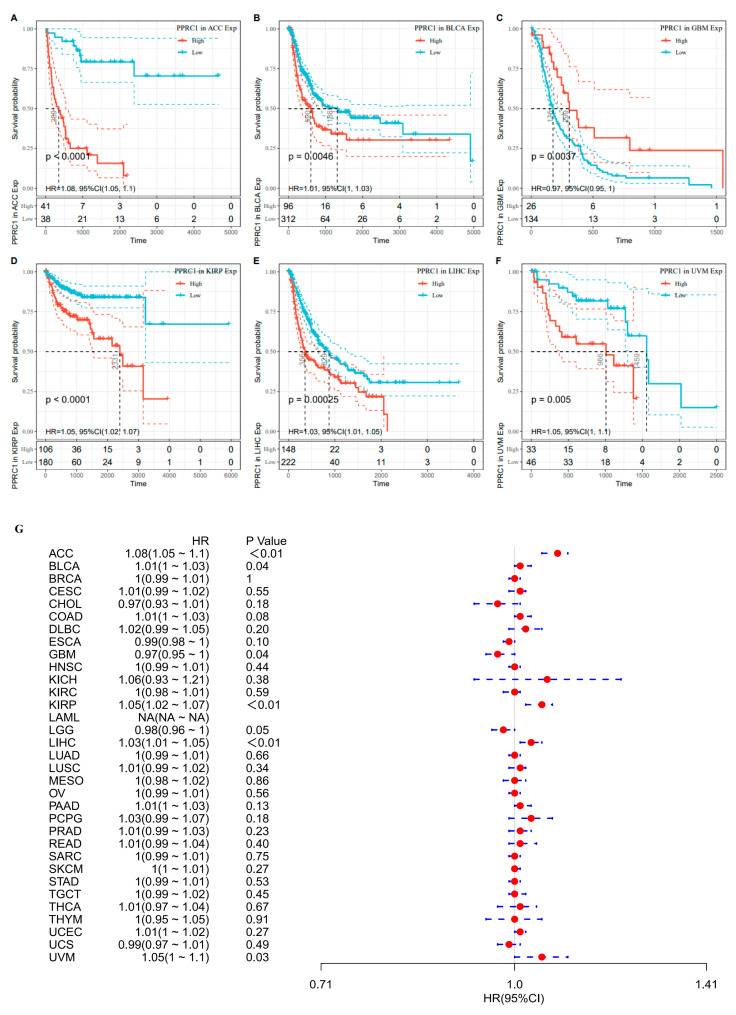
The association of *PPRC1* with PFI. Kaplan–Meier-plotted survival analysis revealed the PFI of ACC (**A**), BLCA (**B**), GBM (**C**), KIRP (**D**), LIHC (**E**), and UVM (**F**). The forest plot shows the hazard ratios of *PPRC1* in pan-cancer (**G**).

**Figure 5 medicina-59-00784-f005:**
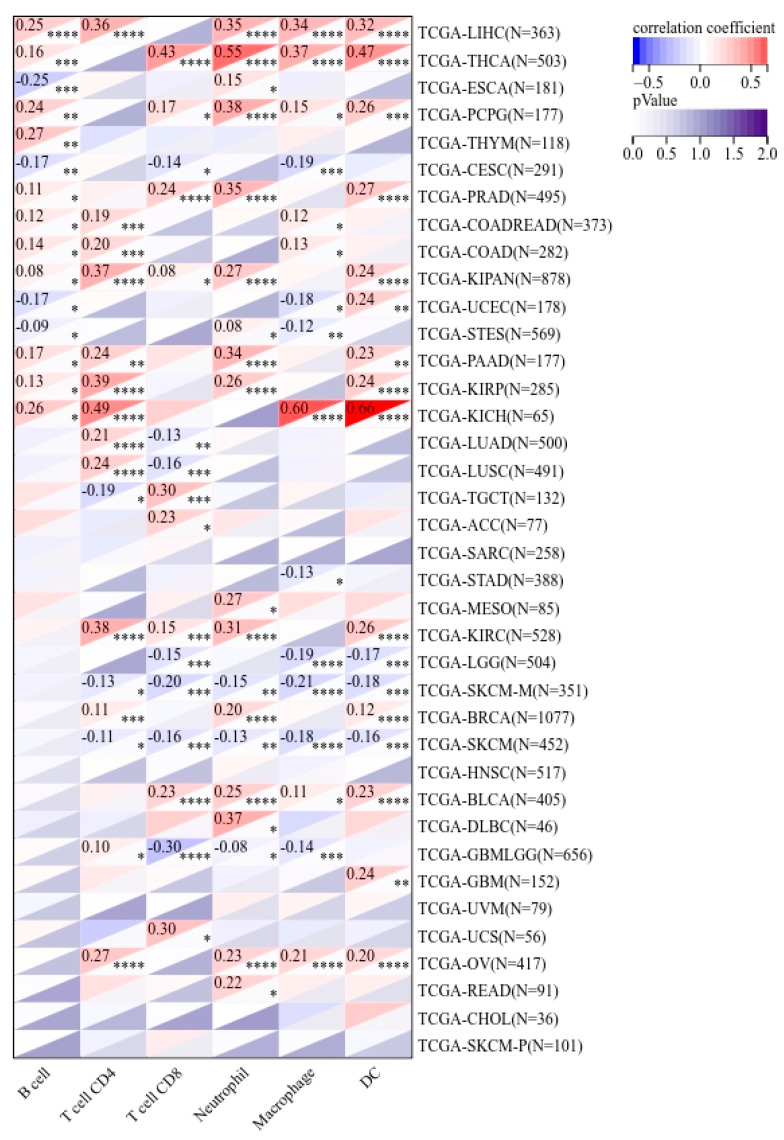
The correlation of *PPRC1* expression and immune cell infiltration. * *p* < 0.05; ** *p* < 0.01; *** *p* < 0.001, **** *p* < 0.0001.

**Figure 6 medicina-59-00784-f006:**
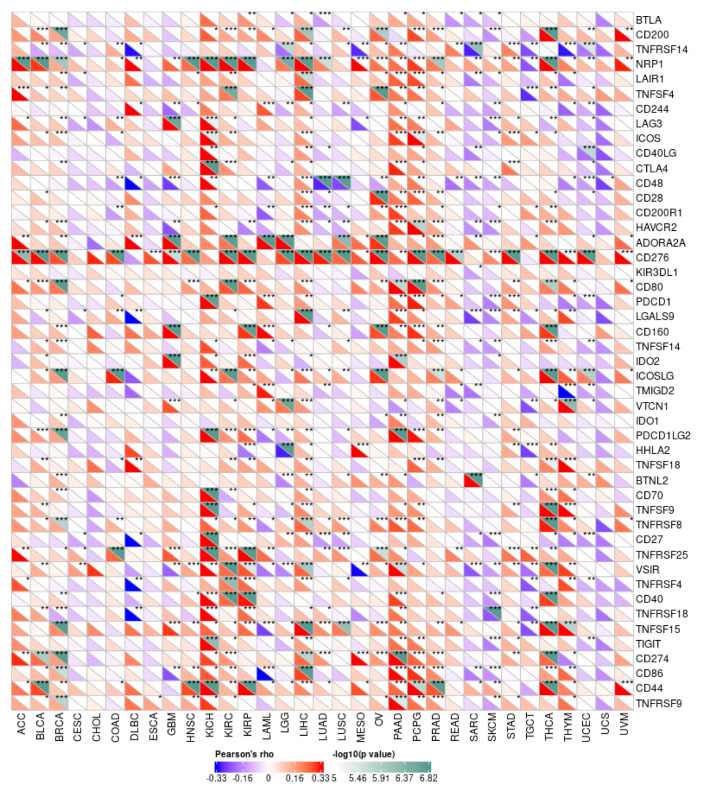
The correlation of PPRC1 expression and tumor immune checkpoints. * *p* < 0.05; ** *p* < 0.01; *** *p* < 0.001.

**Figure 7 medicina-59-00784-f007:**
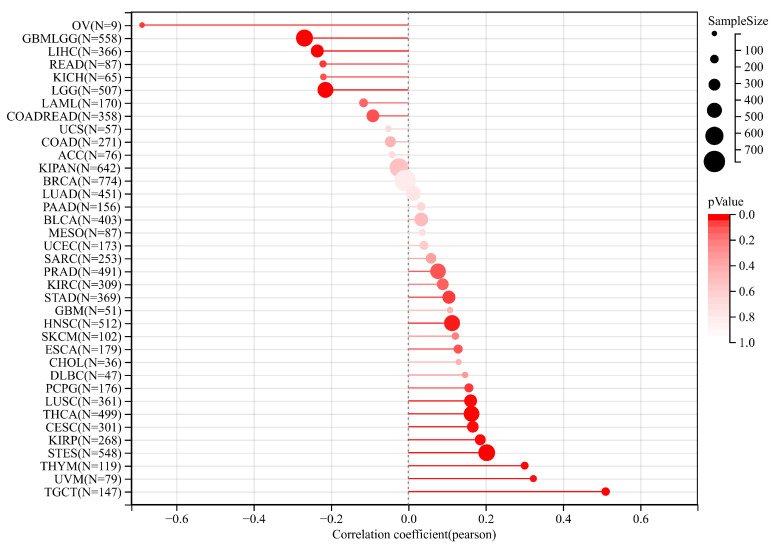
The correlation of PPRC1 expression and tumor-stemness index.

**Table 1 medicina-59-00784-t001:** Correlation of PPRC1 expression and stage in OV and LIHC.

Clinicopathological Characteristics	Ovarian Cancer (Overall Survival, *n* = 1657)			Liver Cancer (Overall Survival, *n* = 364)		
	N	HR	*p*-Value	N	HR	*p*-Value
Stage						
1	107	2.51 (0.75–8.34)	0.12	170	0.74 (0.4–1.37)	0.34
2	61	6.21 (1.96–19.72)	0.0004	83	0.48 (0.21–1.07)	0.065
3	1044	1.11 (0.94–1.31)	0.22	83	2.22 (1.12–4.4)	0.02
4	176	1.7 (1.1–2.62)	0.015	4	Ns	Ns

OV: Ovarian serous cystadenocarcinoma; LIHC: Liver hepatocellular carcinoma.

## Data Availability

All data generated or analyzed during this study are included within the article.

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
