# Peer review of "Pan-Cancer Analysis Reveals PPRC1 as a Novel Prognostic Biomarker in Ovarian Cancer and Hepatocellular Carcinoma"

_medicina, 2023, doi:10.3390/medicina59040784_

Round 1
Reviewer 1 Report
The authors utilize cancer cell line database (CCLE), normal cell database (GTEx) and TCGA (mainly consisting of cancerous tissues) to investigate the prognostic potentials of PPRC1 and to explore the correlations between PPRC1 and other important factors (tumor stemness index, immune celll infiltrations and immune checkpoint expression).
Major issues:
1> Some simple validation experiments can be done to support your conclusions, such as qPCR and WB.
2> IHC staining can be used to validate associations between clinical stages and PPRC1 protein expression level.
Minor issues:
1> In figure 4A to 4F, p values can be moved to top right (so that p values will not be partially covered by dashed lines) and grids of pictures can be removed via ggplot2 theme function.
2> For figure 1A to 1C, the size of texts is too small to be visualized and I recommend that you can adopt circos-like plot to plot them.
Author Response
1)Some simple validation experiments can be done to support your conclusions, such as qPCR and WB.
Response: Thank you for your professional comment. We do agree that some experimental validation can well support the conclusions of this study. However, our department is currently focused on the prognostic and immune roles of PPRC1, and we cannot collect enough samples in a short time to verify the expression level of PPRC1. In addition, the combined application of multiple databases and the expression analysis results of PPRC1 at the cell and tissue levels can confirm the expression of PPRC1 in pan-cancer to a certain extent, which is similar to many other pan-cancer analysis (PMID:34346294, 35069733).
2)IHC staining can be used to validate associations between clinical stages and PPRC1 protein expression level.
Response: Thank you for your professional comment. Based on the high specificity of binding between antibodies and antigens, immunohistochemistry can reveal the relative distribution and abundance of proteins. The Human Protein Atlas (HPA) (https://www.proteinatlas.org) is the largest and most comprehensive human tissue cell protein spatial distribution database, with enormous capabilities and potential application prospects. We used HPA database to observe the differences in PPRC1 expression between normal tissues and ovarian cancer, liver cancer. IHC results showed that the protein expression of PPRC1 in ovarian cancer and liver cancer tissues were up-regulated compared with normal tissues. These results confirmed our findings.
Minor issues:
- In figure 4A to 4F, p values can be moved to top right (so that p values will not be partially covered by dashed lines) and grids of pictures can be removed via ggplot2 theme function.
Response: Thank you for your professional comment. Although deleting the picture grid would be more exquisite, the grid in this figure is mainly to make it easier to visualize the median PFI survival time points of PPRC1 with ACC (A), BLCA (B), GBM (C), KIRP (D), LIHC (E) and UVM (F).
2> For figure 1A to 1C, the size of texts is too small to be visualized and I recommend that you can adopt circos-like plot to plot them.
Response: Thank you for your professional comment. I am very grateful to the reviewer for the proposed changes, but this way of expression in this paper is convenient to visually compare the differences of PPRC1 between normal and abnormal tumor tissues in different databases of GTEx and TCGA.

Reviewer 2 Report
This is a well written pan cancer analysis about PPRC1 that regulates mitochondrial biogenesis and oxidative phosphorylation. The analysis is based on comprehensive analyses of various databases ( GTEx, CCLE, TCGA and TIMER) allowing expression analysis in tumor tissues and adjacent normal tissues in relation to clinical data. PPRC1 could be ascribed a prognostic value for adverse outcome as indicated by overall survival analyses and calculation of Hazard-ratios particularly in ovarian cancer and hepatocellular carcinoma.
Mechanistically, PPRC1 levels could be related to tumor immune cell infiltration, immune checkpoints and tumor stemness index.
The statistical and bioinformatics analyses apparently are well described and presented. The link between PPRC1 and tumor immunologic parameters is of strong interest with innovative character.
Author Response
Thanks to the reviewers for their pertinent review comments!
Reviewer 3 Report
The paper evaluated the expression of pancreatic cancer bio marker PPRC1 and correlate its role in the prognosis of multiple cancers. From their statistical analysis and rigorous database search, the authors were able to find its potential as an early prognosis biomarker in ovarian and Hepatocellular carcinoma.
From the science point of view, please view my comments/questions
- Did the authors check whether the sequence of PPRC1 in different cancers is intact, which means not mutated or indels? Especially in the focused cancers?
- Also, does the referred database contains the PPRC1 expression at different stages of the cancer? Because that can open up a lot of varibles moving in to the clinic.
- Did the authors check if there are any other marker positively/negatively regulated in pan cancer? This would strengthen the detection strategy for further experiments.
Overall, I appreciate the way the authors structured the discussion bringing the immune therapy and how the different types of immune cells act in a tumor micro environment. It would be appropriate if you include an introduction to the different immune cells and their brief function. Therefore, I believe it is a significant finding to the in vitro, in vivo and clinical communities for further research. Hence I suggest the manuscript can be accepted with minor revisions.
Author Response
1)Did the authors check whether the sequence of PPRC1 in different cancers is intact, which means not mutated or indels? Especially in the focused cancers?
Response: Thank you for your professional comment. In TIMER database, we investigate the correlation between the PPRC1 expression and mutation status. The results showed that the tumors with the top three highest frequency of PPRC1 mutations were UCE, SKCM and COAD. Whereas, the frequency of PPRC1 mutations in OV and LIHC were 0.7% and 0.5% respectively.(See attachment for pictures)
2)Also, does the referred database contains the PPRC1 expression at different stages of the cancer? Because that can open up a lot of varibles moving in to the clinic.
Response: Thank you for your professional comment. Based on Kaplan-Meier plotter database, we detected the relation of PPRC1 expression and different stage of LIHC and OV. In OV, higher expression level of PPRC1 was associated with poor prognosis at stage 2 and 4, while in LIHC, higher expression level of PPRC1 was correlated with poor prognosis at stage 3.
Table 1 Correlation of PPRC1 expression and stage in OV and LIHC
|
Clinicopathological characteristics |
Ovarian cancer (Over survival, n=1657) |
Liver cancer (Over survival, n=364) |
||||
|
N |
HR |
P-value |
N |
HR |
P-value |
|
|
Stage |
|
|
|
|
|
|
|
1 |
107 |
2.51(0.75-8.34) |
0.12 |
170 |
0.74(0.4-1.37) |
0.34 |
|
2 |
61 |
6.21(1.96-19.72) |
0.0004 |
83 |
0.48(0.21-1.07) |
0.065 |
|
3 |
1044 |
1.11(0.94-1.31) |
0.22 |
83 |
2.22(1.12-4.4) |
0.02 |
|
4 |
176 |
1.7(1.1-2.62) |
0.015 |
4 |
Ns |
Ns |
3)Did the authors check if there are any other marker positively/negatively regulated in pan cancer? This would strengthen the detection strategy for further experiments.
Response: Thank you for your professional comment. The possible presence of other positive or negative regulatory markers in pan-cancer cannot be excluded. However, in this study, we focus on investigating the prognostic and immune roles of PPRC1 in pan-cancer. It is well known that tumorigenesis is closely associated with oxidative stress, metabolic reprogramming and regulation of mitochondrial energy metabolism. Whereas mitochondrial synthesis is mainly activated by the gamma coactivator (PGC) family of peroxisome proliferator-activated receptors, recent reports in the literature also suggest that PPRC1 is not only involved in the regulation of glioblastoma, but also contributes to the diagnosis and understanding of the pathogenesis of pleomorphic low-grade neurons in young adults
sai HF, Chang YC, Li CH, Chan MH, Chen CL, Tsai WC, Hsiao M: Type V collagen alpha 1 chain promotes the malignancy of glioblastoma through PPRC 1-ESM1 axis activation and extracellular matrix remodeling. Cell death discovery 2021, 7(1):313.
Yu G, Wang LG, Han Y, He QY: clusterProfiler: an R package for comparing biological themes among gene clusters. Omics : a journal of integrative biology 2012, 16(5):284-287.

Round 2
Reviewer 1 Report
It can be accepted.